# MULTIMODAL FUNCTION VECTORS FOR SPATIAL RELATIONS

## ABSTRACT

Large Multimodal Models (LMMs) demonstrate impressive in-context learning abilities from limited multimodal demonstrations, yet the internal mechanisms supporting such task learning remain opaque. Building on prior work of large language models, we show that a small subset of attention heads in two vision–language model, OpenFlamingo and Qwen3-VL, is responsible for transmitting representations of spatial relations. The activations of these attention heads, termed *function vectors*, can be extracted and manipulated to alter an LMM's performance on relational tasks. First, using both synthetic and real image datasets, we apply causal mediation analysis to identify attention heads that strongly influence relational predictions, and extract multimodal function vectors that improve zero-shot accuracy at inference time. We further demonstrate that these multimodal function vectors can be fine-tuned with a modest amount of training data, while keeping LMM parameters frozen, to significantly outperform in-context learning baselines. Finally, we show that relation-specific function vectors can be linearly combined to solve analogy problems involving novel and untrained spatial relations, highlighting the strong generalization ability of this approach. Our results show that LMMs encode spatial relational knowledge within localized internal structures, which can be systematically extracted and optimized, thereby advancing our understanding of model modularity and enhancing control over relational reasoning in LMMs.

## 1 INTRODUCTION

Imagine you look at a picture of a kitchen. Without identifying relations between objects, the visual system might perceive a disconnected list: fridge, boy, cabinet, sink, window. However, with relational representations, the system provides a much richer description: a boy is *opening* a fridge that is *next to* a cabinet; the cabinet is *besides* a window, which is *above* the sink. Simply reading this description with relational context makes it far easier to imagine the scene as shown in Figure 1. This thought experiment highlights the critical role that relational representations play in perception, enabling us to organize and make sense of the world by interpreting it as interconnected, meaningful scenes, and also to form a "language of vision" to communicate with cognitive systems (Cavanagh, 2021).

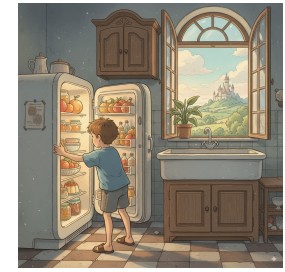

Figure 1: Relational representations enrich perception: rather than a disconnected list of objects, relations (e.g., the boy *opening* the fridge *next to* the cabinet) provide a structured, meaningful description.

Although the importance of relational context is evident in shaping a "language of vision," it remains a difficult challenge because "relations themselves cast no light onto our eyes" (Hafri & Firestone, 2021). In other words, no pixels in an image signal visual relations. However, recent advances in the mechanistic interpretability of large language models (LLMs) suggest that in-context learning can offer a promising pathway for distilling relational knowledge from pre-trained models. In particular, one key line of research focuses on inference-time modification of model activations to make task representations explicit (Turner et al., 2023). Here, we particularly focus on the approch of *function vectors* (FVs) (Todd et al., 2024). Function vectors were recently developed as a means to extract compact

representations of a task from the hidden states of LLMs. By averaging activations from a small number of attention heads across a set of consistent demonstrations, researchers have shown that it is possible to define a task-specific vector. The extracted function vector can be inserted into a model's hidden layers and produce the intended behavior for a task even in the absence of any demonstrations. These vectors effectively summarize the task's input-output mapping and can be reused, combined, or fine-tuned for new contexts (Jorgensen et al., 2023; Yin et al., 2024; Park et al., 2023).

Despite the promise of function vectors in LLMs, their extension to multimodal settings remains at an early stage. LMMs such as Flamingo (Alayrac et al., 2022), BLIP (Li et al., 2022), LLaVA (Liu et al., 2023b;a; 2024), and Qwen-VLBai et al. (2023); Wang et al. (2024); Bai et al. (2025) introduce additional complexity due to the fusion of high-dimensional visual features with natural language, posing unique challenges for representation analysis. Recent work has successfully identified task vectors in pre-trained vision–language models for visual prompting (Hojel et al., 2024; Huang et al., 2024), e.g., modifying display styles or naming objects. Yet, the function vector approach has not been explored for extracting and manipulating visual relations in images.

This paper investigates whether the approach of function vectors can be effectively extended to large multimodal models (LMMs) to support the extraction of relational knowledge in images. Specifically, we ask whether multimodal function vectors can be extracted from the internal representations of LMMs in a way that encodes spatial relations in a compact and causally meaningful form. We further explore how architectural factors, such as the selection of attention heads and the choice of injection layer, influence the effectiveness of function vector interventions. Next, we examine whether these multimodal function vectors can be fine-tuned with a modest amount of training data consisting of object pairs instantiating the same relations, while keeping model parameters frozen. We will compare performance of fine-tuned function vectors with LMM's in-context learning baselines. Finally, inspired by the linear representation hypothesis (Park et al., 2023) in transformer-based models, we hypothesize that relation-specific function vectors can be linearly combined to represent untrained relations. We test this idea using one-shot analogy problems to examine generalization of this approach.

## 2 RELATED WORK

### 2.1 IN-CONTEXT LEARNING AND FUNCTION VECTORS IN LARGE LANGUAGE MOELS

Large Language Models show impressive in-context learning ability (ICL), which can be viewed as implicit meta-learning: attention dynamics approximate gradient descent or Bayesian inference (Brown et al., 2020; Garg et al., 2022; Xie et al., 2021; Akyürek et al., 2023). Empirical work highlights that label words (Wang et al., 2023a), label noise (Wang et al., 2023b), and topical coherence (Wang et al., 2023c) can influence prediction performance.

Recent work used in-context learning to show that transformer-based Large Language Models use local structures to encode tasks using compact, causally meaningful representations (Hendel et al., 2023). For example, (Olsson et al., 2022) identified "induction heads" enabling few-shot generalization of copying token patterns forward in a sequence. Built on the idea of induction heads, Todd and colleagues developed the function vector (FV) framework (Todd et al., 2024) to show that a small subset of mid-layer attention heads encodes the input-output mapping implied by in-context examples. Hence, the average activations of these selected attention heads can yield a single function vector to capture task representations. Intervening on the language model with function vectors reproduces task behavior without demonstrations. In this paper, we extend this paradigm to multimodal models, testing whether vision-language systems such as Flamingo (Alayrac et al., 2022) also encode multimodal tasks as function vectors.

### 2.2 MECHANISTIC INTERPRETABILITY IN MULTIMODAL MODELS

Mechanistic interpretability has uncovered circuits and features that support model behavior in transformer-based Large Language Models. For example, Variengien & Winsor (2023) decomposed question-answer problems into query and retrieval stages to reveal modularity in transformers. (Wang et al., 2022a) mapped a pronoun resolution circuit in GPT-2, while "skill neurons" (Wang et al., 2022b) and "knowledge neurons" (Meng et al., 2022) revealed latent units causally tied to task

execution and factual recall. Tools like the Tuned Lens (Belrose et al., 2023) and large-scale feature maps (Anthropic, 2024) further demonstrate structured internal organization.

Extending mechanistic interpretability to Large Multimodal Models is challenging due to fused vision–language streams (Dang et al., 2024). However, progress has been made. Causal tracing in BLIP (Palit et al., 2023) found late-stage integration, while automatic circuit discovery isolates concept-specific subnetworks (Rajaram et al., 2024). Meanwhile, Visual Task Vectors have been discovered for visual prompting tasks (Hojel et al., 2024). Multimodal Task Vectors (MTV) show that task information can be summarized into a reusable vector (Huang et al., 2024). Our work derives function vectors via causal mediation analysis and fine-tuning, enabling manipulation of relational knowledge and generalization to solving analogy problems with untrained relations.

## 3 METHOD

### 3.1 DATASETS

We use two multimodal datasets to test the models, one with synthetic images and the other with realistic images. Full construction details are provided in the Supplementary Material A.1

*Synthetic image dataset.* We constructed a synthetic image dataset using 42 object cutouts from the Big and Small Objects dataset (Konkle & Oliva, 2012). Each image includes six objects arranged to instantiate specific spatial relations. Four relations are considered: *above*, *below*, *left of*, and *right of*. One object is designated as the reference object, which consistently serves as the **query object** in the relational reasoning task. Each image includes a centrally placed reference object, four relational objects corresponding to the target spatial relations, and one additional object positioned at least 300 pixels away from all others. We selected 32 objects among the 42 objects. We used in-context learning to make sure that OpenFlamingo and Qwen3-VL can correctly identify the object name from images. We also used Chat-GPT to further confirm the recognition. A total of 7000 images by placing objects in different locations are generated for the synthetic dataset. These were divided into four subsets: (1) 4000 images for extracting function vectors, (2) 1000 images for fine-tuning function vectors, (3) 1000 images for evaluating generalization, and (4) 1000 images for a relation generalization test dataset containing four novel spatial relations not present in training: *above left*, *above right*, *below left*, and *below right*. We then used the held-out 10 objects to generate another 1000 images as an object generalization test set.

*Real image dataset: GQA.* For more realistic settings, we constructed a dataset using the GQA (Hudson & Manning, 2019), which consists of real-world images annotated with detailed scene graphs supporting visual reasoning and question answering. From the 113K images in GQA, we selected 4,226 images using strict criteria designed to target relational tasks. The dataset includes 7 spatial relations, including *above, behind, below, in front of, next to, left of, right of*. See detailed criteria in Appendix A.1.2. We divided the dataset into two subsets: a training set with 2,113 images, used for function vector extraction and fine-tuning, and a test set with the other 2,113 images, used exclusively for evaluation with zero-shot tasks. From each subset, we sampled 1000 tasks per relation, where each task comprises four context images and one query image, with object pairs instantiating the relation randomly selected.

### 3.2 RELATION TASK

To evaluate how the large multimodal models represent spatial relations, we designed a 4-shot in-context learning task (ICL) to test three multimodal models, OpenFlamingo-4B (Awadalla et al., 2023), LLaVA-OneVision-1.5-4B-InstructAn et al. (2025), and Qwen3-VL-4B-instruct (Team, 2025; Bai et al., 2025). Each multimodal prompt consisted of four context images and one query image, accompanied with text inputs. In the in-context demonstrations, four examples consistently include a specific spatial relation (e.g., *above*) between a query object (Q) and its corresponding answer object (A). Followng these demonstrations, a query image with the text label of a query object is presented, and the model must infer the linguistic label of an object that instantiates the correct spatial relation with the query object. See an illusation of the relation task in the in-context learning settings in the top panel of Figure 2.

OpenFlamingo (Awadalla et al., 2023) integrates a frozen CLIP vision encoder with a RedPajama-INCITE language model using interleaved cross-attention layers inserted every two transformer blocks[1]. LLaVA-OneVision-1.5 (An et al., 2025) employs a unified vision–language architecture that pairs a vision transformer with a LLaMA-based backbone to support image understanding, grounding, and multimodal dialogue[2]. Qwen3-VL (Team, 2025; Bai et al., 2025) also follows a unified multimodal design, combining a vision encoder with a Qwen language model and enhancing cross-modal alignment through Interleaved-MRoPE for long-horizon spatial–temporal reasoning and a hierarchical DeepStack fusion mechanism[3].

The relation task remains a challenging task for large multimodal models. We tested the in-context learning performance on the three multimodal models, including Open-Flamingo-4B, LLaVA-OneVision-1.5-4B-Instruct, and Qwen3-VL-4B-Instruct. The results are reported in Table 1 in Appendix A.1.2. We found that Open-Flamingo's performance is consistently low across all shot settings, reaching at most 9.7% accuracy on the Synthetic dataset and 19.0% on GQA, indicating that it struggles to benefit meaningfully from in-context examples. LLaVA-OneVision shows moderate gains with increasing shots, but its overall accuracy remains limited, especially on Synthetic. Although Qwen3-VL demonstrates the strongest in-context learning behavior among the three models, improving from 19.5% (0-shot) to 26.8% (4-shot) on Synthetic and from 21.6% to 31.6% on GQA, its performance is still far below human levels, for whom this relation task is essentially trivial. This gap highlights the continuing difficulty large multimodal models face in acquiring systematic relation concepts even when provided with demonstrations.

### 3.3 EXTRACTING FUNCTION VECTORS IN MULTIMODAL CONTEXTS

We extracted function vectors from both the OpenFlamingo model (Awadalla et al., 2023) and the Qwen3-VL model (Team, 2025). For each model, we focus on layers within the language module, where multimodal information from the vision encoders is integrated. Our aim is to capture the internal representations associated with spatial relations in the input images. In particular, we investigate whether function vectors (FVs) corresponding to these relations can be explicitly identified and then causally manipulated to influence model behavior on multimodal relation tasks.

#### 3.3.1 FORMULATION

Let $f$ denote a vision-language transformer model and $t$ denote a relation task (e.g., identifying the object that is *right of* or *above* a query object). For each task $t$, we construct ICL prompts $p_i^t \in P_t$ that consist of a sequence of image-text examples. Each example encodes a pair $(x_k, y_k)$ in the format:

```
<image>Q:x_k.  A:y_k.  <|endofchunk|>
```

A complete prompt includes several such in-context demonstration examples followed by a query. For a task prompt $p_i^t$ with $n$ context pairs and a query input $x_q$, the structure is:

$$p_i^t = \text{<image>Q:}x_1.\quad \text{A:}y_1.\quad \text{<|endofchunk|>}$$
$$\cdots$$
$$\text{<image>Q:}x_n.\quad \text{A:}y_n.\quad \text{<|endofchunk|>}$$
$$\text{<image>Q:}x_q.\quad \text{A:}$$

The model is expected to infer the correct answer $y_q$ based on the context and query object.

#### 3.3.2 CAUSAL MEDIATION ANALYSIS

Let $a_{\ell j}(p_i^t)$ represent the activation of the $j$-th attention head at layer $\ell$ when processing prompt $p_i^t$. For each attention head, we compute the relation-specific average activations, mean of task-conditioned activations across all prompts for a specific relation $t$ as:

$$\bar{a}_{\ell j}^t = \frac{1}{|P_t|} \sum_{p_i^t \in P_t} a_{\ell j}(p_i^t) \tag{1}$$

---

[1]Initialized from `openflamingo/OpenFlamingo-4B-vitl-rpj3b-langinstruct`.

[2]Initialized from `lmms-lab/LLaVA-OneVision-1.5-4B-Instruct`.

[3]Initialized from `Qwen/Qwen3-VL-4B-Instruct`.

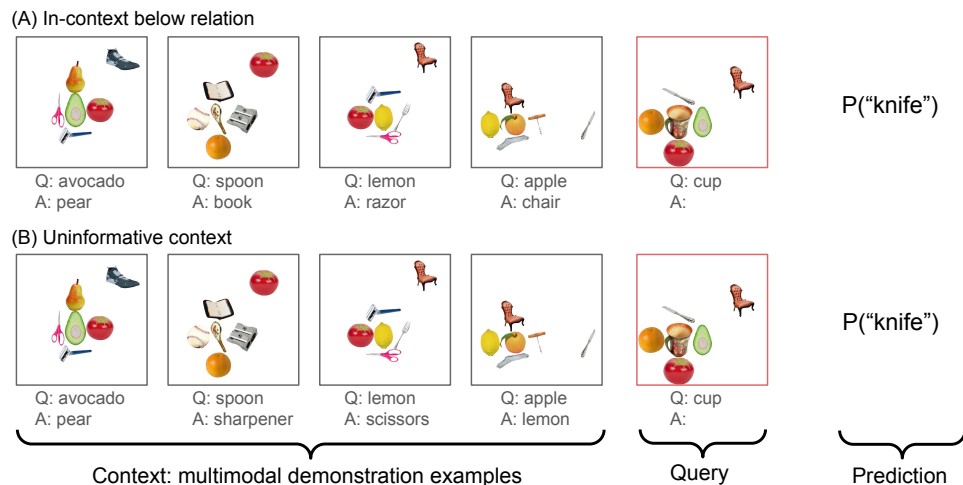

Figure 2: **Example 4-shot in-context learning (ICL) prompts for relation understanding.** Each prompt includes four demonstrations followed by a query. We compare the model's performance in a consistent relational setting (A) versus a perturbed setting (B) to isolate components responsible for relational inference.

To assess the causal influence of attention heads, we construct perturbed prompts with uninformative context $\tilde{p}_i^t \in \tilde{P}_t$. An uninformative context is generated by pairing the reference object with a randomly chosen object $x_k$ that does not exhibit the target relation in the image $\tilde{y}_k$. To prevent the in-context demonstrations from being biased toward any particular relation, the sampled object labels $\tilde{y}_k$ are selected such that each of the four relation types—above, below, left of, and right of—appears exactly once across the four image-text pairs in each perturbed prompt. See Figure 2 bottom panel for an example.

We then run the model on perturbed prompts with uninformative context twice: once with original activations and once with the attention head activation $a_{\ell j}$ replaced by the relation-specific mean activations computed from the in-context learning $\bar{a}_{\ell j}^t$. The causal indirect effect (CIE) of an attention head $a_{\ell j}$ is defined as the difference of prediction probability between the two rans.

To quantify the overall contribution of an attention head in processing a specific relation, we compute its *average indirect effect* (AIE) as defined in Todd et al. (2024). This metric reflects the mean increase in the model's probability of generating the correct object label when the activation of attention head $a_{\ell j}$ is replaced by its relation-secific mean activations $\bar{a}_{\ell j}^t$ for perturbed prompts. The heads with the highest AIE scores are identified as the most causally influential for task execution and are grouped into the set $\mathcal{A}_t$.

We define the function vector $\mathbf{v}_t \in \mathbb{R}^d$ for a specific relation task $t$ as the sum of mean activations from the selected top heads with high AIE in $\mathcal{A}_t$:

$$\mathbf{v}_t = \sum_{a_{\ell j}^t \in \mathcal{A}_t} \bar{a}_{\ell j}^t \tag{2}$$

### 3.3.3 ZERO-SHOT INTERVENTION WITH RELATION-SPECIFIC FUNCTION VECTORS

To evaluate whether the relation-specific function vector $\mathbf{v}_t$ captures a transferable and causally meaningful representation for performing a relation task, we perform an intervention in a zero-shot setting, in which a prompt contains no prior in-context demonstrations of the task. Let $\tilde{p}_i^{\emptyset}$ denote a zero-shot prompt containing only the query image and query object label, without any in-context examples. We intervene on the model's hidden state at a selected layer $\ell$ by adding the relation-specific function vector $\mathbf{v}_t$. Specifically, we modify the hidden representation $\mathbf{h}^{(\ell)}(\tilde{p}_i^{\emptyset})$ at the final token position as:

$$\mathbf{h}^{(\ell)}(\tilde{p}_i^{\emptyset}) \leftarrow \mathbf{h}^{(\ell)}(\tilde{p}_i^{\emptyset}) + \mathbf{v}_t. \tag{3}$$

We then evaluate whether the model produces the correct object label $y_q$ in response to the query to instantiate the intended relation. Model performance is evaluated using top-1 prediction accuracy, defined as the proportion of test queries where the model's highest-ranked output correctly predicts the first token of the object label corresponding to the intended spatial relation. If the intervention of adding a relation-specific function vector during inference increases accuracy compared to the zero-shot baseline, we interpret this as evidence that $\mathbf{v}_t$ embeds the intended relational knowledge and can causally trigger task execution even without in-context demonstrations.

### 3.3.4 FINE-TUNING FUNCTION VECTORS ON ZERO-SHOT PROMPTS

Next, we introduce a fast-learning component to fine-tune relation-specific function vector $\mathbf{v}_t$ using a held-out set of zero-shot multimodal examples, freezing all model parameters and updating only relation-specific function vectors.

Let the zero-shot training set be denoted by $\mathcal{D}_t^{\text{train}} = \{(\tilde{p}_i^\emptyset, y_q^i)\}_{i=1}^N$, where $\tilde{p}_i^\emptyset$ is a prompt containing only the query image and query object label, and $y_q^i$ is the correct object label indicating the relation to the query object. We then optimize $\mathbf{v}_t \in \mathbb{R}^d$ to increase the model's likelihood of producing the correct answer. The training objective is the negative log-likelihood over the training set:

$$\mathcal{L}(\mathbf{v}_t) = -\frac{1}{N} \sum_{i=1}^N \log f(\tilde{p}_i^\emptyset \mid \mathbf{h}^{(\ell)} + \mathbf{v}_t)[y_q^i] \tag{4}$$

Note that the backbone model $f$ remains completely frozen during this fine-tuning procedure; only the function vector is updated.

The fine-tuning procedure is conducted on a dedicated training set of 1000 zero-shot examples in the synthetic dataset or the 2,113 training images in the real image dataset, respectively, as described in Section 3.1. Each example consists of a single query image and a query object name, without any in-context demonstrations. During training, the relation-specific function vector is injected into the hidden representation at a selected layer $\ell$ (layer 19 for synthetic dataset, and layer 8 for real-image GQA dataset), specifically at the final token position, and is optimized to increase the model's probability of generating the correct lable of object that couples with the query object to instantiate a specific relation. The fine-tuning process is initialized with the extracted relation-specific function vector from the causal mediation analysis, and proceeds for 20 epochs using the Adam optimizer with a learning rate of 0.001 and a cosine annealing learning rate schedule.

We evaluate generalization performance separately on the held-out test sets of the two datasets. The synthetic test set includes 1,000 zero-shot examples, while the GQA test set corresponds to the designated split described above. In both cases, the test data are entirely disjoint from the extraction and training sets.

### 3.3.5 COMPOSITE FUNCTION VECTORS FOR ONE-SHOT ANALOGY TASK

One characteristic of explicit relational knowledge is that the knowledge can be actively manipulated to guide the inference process. Here, we use relation-specific function vectors as a basis to compute the representation of other spatial relations that are not included in the training set. This idea is consistent with the "linear representation hypothesis" that high-level concepts can be represented linearly in a model's internal representation space (e.g. (Mikolov et al., 2013; Elhage et al., 2022; Park et al., 2023)).

We develop a two-step procedure for solving one-shot analogy problems involving these untrained spatial relations. (1) *Compute a composite function vector from a source analogy.* Given a source object pair $(x_1, y_1)$ in an image, we compute a composite function vector as a weighted sum of relation-specific function vectors. The weight assigned to each function vector $\mathbf{v}_t$ is proportional to the model's probability of predicting $y_1$ given $x_1$ and $\mathbf{v}_t$:

$$w_t = \frac{P(y_1 \mid x_1, \mathbf{v}_t)}{\sum_{t'} P(y_1 \mid x_1, \mathbf{v}_{t'})}. \tag{5}$$

The resulting composite function vector is then defined as:

$$\mathbf{v}_{\text{composite}} = \sum_t w_t \mathbf{v}_t. \tag{6}$$

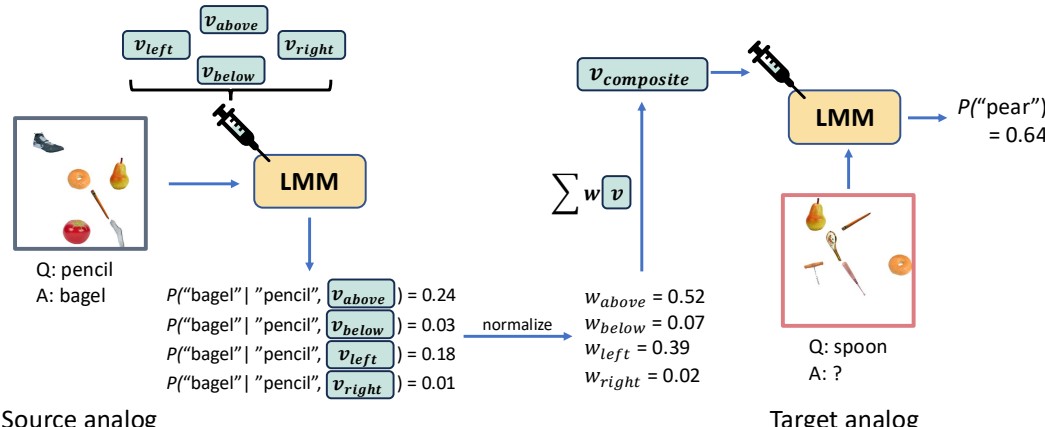

Figure 3: Illustration of the composite function vector approach for one-shot analogy tasks. In the source analogy, relation-specific function vectors $\mathbf{v}_t$ are injected into the model to compute prediction probabilities for the target object $y_1$ given the reference object $x_1$. These probabilities define the weights $w_t$ used to form a composite function vector $\mathbf{v}_{\text{composite}}$ as a weighted sum of $\mathbf{v}_t$. The resulting vector is then transferred to guide inference in the target analogy.

(2) *Complete the target analogy.* We inject the composite function vector $\mathbf{v}_{\text{composite}}$ into the model to perform zero-shot inference on the target analogy . This transfer allows the model to generalize relational knowledge instantiated in the source pair $(x_1, y_1)$ to the new target setting.

Figure 3 provides an illustration of this process. Note that the composite function vector is constructed for a particular source object pair and image. It encodes the relation instantiated between these objects in a source and transfers that relational knowledge to guide inference in the target analog.

In our setting, zero-shot inference is not well-defined for composite relations, because the model lacks any means to infer the mixture weights in Eq. 5 without observing a source example. Although we refer to this as a "one-shot analogy" task, the source image is not provided to the model at inference time; it is used only to compute the relational weights that define the composite function vector. Thus, the model still performs inference in a zero-shot manner on the target analogy, but the construction of the composite vector inherently requires one analogy example. As a consequence, reporting a zero-shot baseline for composite relations would be misleading, since the task itself fundamentally requires at least one source instance to determine how the base relations should be combined. For completeness, we note that one could impose hand-crafted weights (e.g., $\mathbf{v}_{\text{above-right}} = 0.5\mathbf{v}_{\text{above}} + 0.5\mathbf{v}_{\text{right}}$), but this would remove the analogy component of the task and artificially boosts performance while no longer evaluating compositional generalization.

## 4 EXPERIMENTS

### 4.1 IDENTIFYING CAUSALLY IMPORTANT ATTENTION HEADS FOR SPATIAL RELATIONS

We first compute the Average Indirect Effect (AIE) for each attention head, for each specific spatial relation. This allows us to rank heads by their causal contribution to relational predictions. Figure 4 shows the distribution of AIE scores across all layers and heads for the *above* relation for the OpenFlamingo model and the Qwen3-VL model on the synthetic dataset. The AIE score figures for other relations in synthetic dataset and real image dataset are included in the Supplementary Material Figure 7 and 8. We observe that only a small subset of heads concentrated in intermediate layers exhibit consistently high AIE scores. We select the top 10 attention heads with the highest AIE as the causal subnetwork $\mathcal{A}_t$ for each relation task. The function vector for each relation $\mathbf{v}_t$ is then calculated by averaging the activations from the selected top 10 heads.

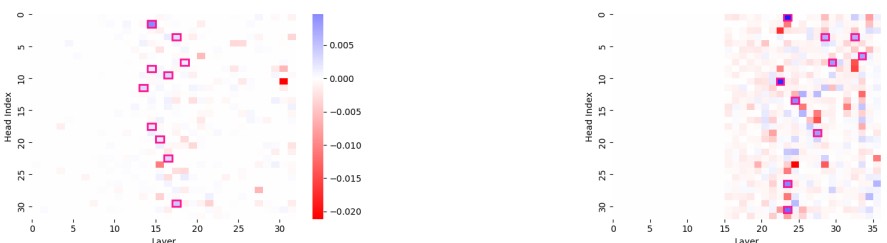

Figure 4: **Average indirect effect (AIE) scores for the *above* relation in OpenFlamingo (left) and Qwen3-VL (right).** Each heatmap displays AIE values for attention heads, indexed by layer and head position. Pink boxes highlight the top 10 attention heads with the strongest causal influence.

## 4.2 EFFECTS OF INTERVENTION LAYER, HEAD SET SIZE, CONTEXT SIZE

We examined how the effectiveness of function vector interventions depends on the injection layer, the number of attention heads used, and the size of the in-context prompt. All detailed results are in Supplementary Material A.4. Below we summarize the main findings for these factors.

*Layer effect.* Zero-shot accuracy peaks when function vectors are injected at intermediate layers (e.g., around layer 19 for synthetic data), while early layers lack sufficient abstraction and late layers are too downstream to support relational reasoning between objects.

*Head set size.* Performance improves rapidly as more top-ranked heads are included, peaks with a small subset (6–12 heads), and then declines as less informative heads introduce noise. This reveals a trade-off: too few heads underrepresent relational knowledge, while too many dilute the signal with irrelevant activations. Across both synthetic and real-image datasets, function vectors built from a sparse, carefully chosen set of attention heads significantly outperform the zero-shot baselines.

*Context size.* Function vector performance remains relatively stable across 2-shot and 4-shot prompts, with only marginal changes at 8-shot. In some cases, longer contexts slightly reduce accuracy, possibly reflecting model capacity limits. These results hold for both earlier models (e.g., OpenFlamingo) and more recent ones (e.g., Qwen3-VL-4B-Instruct). These findings indicate that moderate context is sufficient to obtain robust activations of function vectors, and more context does not necessarily improve performance of function vectors in LMMs.

## 4.3 FINE-TUNING FUNCTION VECTORS FOR ZERO-SHOT RELATION TASKS

To evaluate generalization, we use two separate held-out test sets. For the synthetic dataset, the test set contains 1,000 zero-shot examples. For the real image dataset, the test set includes 2,113 images corresponding to 1000 tasks per relation. Both test sets are fully disjoint from the extraction and training data. Figure 5 reports prediction accuracy for 4 spatial relations in synthetic dataset and 7 relations in real-image GQA dataset. The plots include performance from four setting, (1) the LMM zero-shot baseline, (2) standard LMM 4-shot in-context learning, (3) the initial (untrained) relation-specific function vector based on causal mediation analysis, and (4) the finetuned function vector (FFV).

As shown in Figure 5, for the synthetic datasets, we observe that fine-tuning leads to substantial performance gains in both the OpenFlamingo and the more recent Qwen3-VL model. Here we show the average performance and the per-relation accuracies are included in Appendix Figure 12. For the GQA real-image dataset, we found that fine-tuned FV showed the best performance across seven spatial relations for OpenFlamingo. Finetuned function vectors more than double the accuracy achieved in the zero-shot baseline and outperform both the 4-shot ICL condition and the initial function vector for both models. These findings highlight that function vectors are not only causally meaningful encodings of relation-specific representations, but also flexible and optimizable representations that can be adapted to novel inputs.

For the synthetic dataset, we ran a second simulation using the test set consisting of 10 novel objects never appeared in training. We found that the superior performance of FFVs persisted even with these novel test objects. For OpenFlamingo, the FFV achieved the highest accuracy at 13.6%,

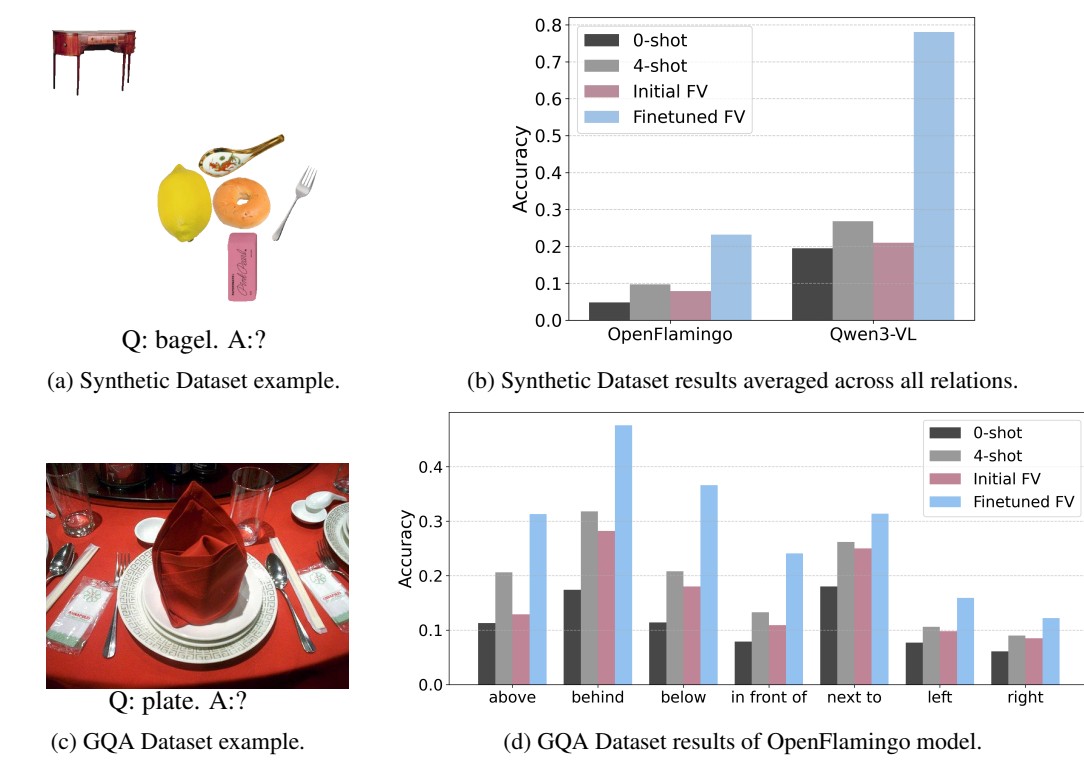

(a) Synthetic Dataset example.

(b) Synthetic Dataset results averaged across all relations.

(c) GQA Dataset example.

(d) GQA Dataset results of OpenFlamingo model.

Figure 5: **Top-1 prediction accuracy of zero-shot relation tasks** for four settings: zero-shot baseline of LMM, four-shot ICL of LMM, initial function vector, and finetuned function vector. Finetuned vectors significantly outperform all baselines on the held-out zero-shot test set.

outperforming both the initial FV (11%) and the baseline model (9.4%). For the Qwen3-VL model, FFVs produced substantial gains, reaching 72.2% compared to 23.3% for the initial FV and 21.3% for the baseline. These results indicate that the FFV model generalizes well to datasets containing novel objects, suggesting that it captures relational knowledge that is independent of the specific entities involved.

## 4.4 COMPOSITE FUNCTION VECTORS FOR ONE-SHOT ANALOGY TASK

We evaluate composite function vectors (CFVs) on one-shot analogy tasks involving untrained spatial relations (*above-left, above-right, below-left, below-right*). The test set contains 1000 one-shot analogy problems. To construct the CFVs, we derive function vectors from four primary spatial relations (*above, below, left-of, right-of*) in the source analogy and combine them through weighted averaging. The resulting CFV is then applied to the target analogy during inference. Model performance with CFVs is compared against three baselines: zero-shot, one-shot and four-shot in-context learning. As shown in Figure 6 (right panel), the CFV on OpenFlamingo model achieved substantial improvements, doubling accuracy from 8.1% in four-shot ICL to 16.8% with CFV. A similar result is also observed for CFV on the Qwen3-VL model, with CFVs improving accuracy from 28.7% in the four-shot ICL condition to 45.1%, outperforming all ICL baselines.

## 5 CONCLUSION

This paper set out to investigate whether the concept of function vectors could be extended from language-only transformer models to large multimodal models (LMMs), with a focus on relational reasoning tasks. By targeting vision-language models, including OpenFlamingo and Qwen3-VL, we developed a framework to extract, analyze, and manipulate function vectors derived from structured in-context learning prompts.

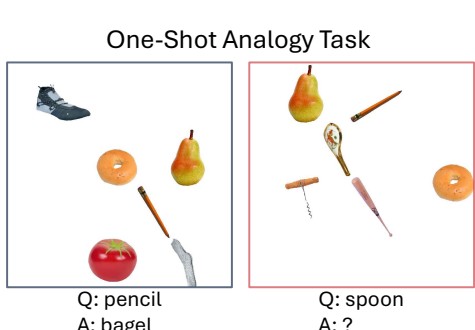

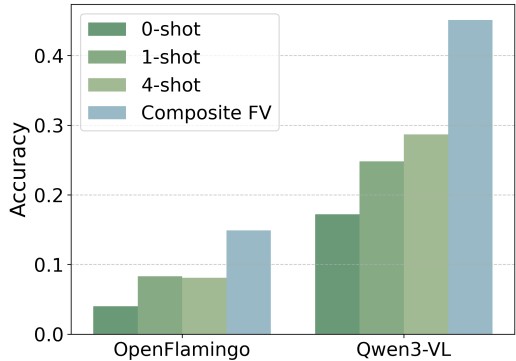

Figure 6: **Top-1 prediction accuracy of one-shot analogy tasks** for composite function vectors (CFVs) involving untrained spatial relations (*above-left*, *above-right*, *below-left*, *below-right*). The CFV model outperformed baseline in-context learning models.

The experimental results demonstrate that function vectors can indeed be extracted from the activations of a sparse subset of attention heads in LMMs and that these vectors retain causal influence over the model's output. Specifically, injecting function vectors into zero-shot prompts significantly increased the model's ability to make correct relational predictions. This confirms that the extracted vectors encode relational knowledge beyond superficial memorization of context. Furthermore, after fine-tuned on zero-shot examples, these vectors yielded substantial gains in performance, surpassing the few-shot in-context learning baseline. These findings validate function vectors as flexible and transferable modules that can be used to control and enhance reasoning in LMMs. Importantly, these relation-specific function vectors can be linearly combined to represent previously untrained relations. The composite function vectors demonstrated significant improvements over LMM in-context learning baselines in solving one-shot analogy problems using untrained relations. This serves as a proof of concept demonstrating the potential of this approach to generalize to out-of-distribution relations.

While this study presents promising results, several limitations must be acknowledged. First, the scope of relational tasks investigated in this work is restricted to a small set of spatial relations (e.g., above, next to). Although this controlled setting allows for precise causal analysis, it does not capture the full richness or diversity of visual relations required in real-world multimodal tasks. Future work should explore whether the multimodal function vector framework generalizes to larger, more advanced multimodal architectures with different training regimes and fusion mechanisms. Extending the analysis to a broader range of relational categories—including physical, agentic, and social relations—would also test the flexibility of the approach. Second, applying CFVs requires contextual demonstrations to determine what relational function vectors should be injected. Therefore, the minimal setting in which this method is applicable is one-shot in-context learning focused on relations between entities. Future work is needed to systematically examine the interplay between relation knowledge and context in reasoning tasks.

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

# A  SUPPLEMENTAL MATERIALS

## A.1  DATASETS

### A.1.1  SYNTHETIC IMAGE DATASET

We constructed a synthetic image dataset using object cutouts from the Big and Small Objects dataset (Konkle & Oliva, 2012), which contains real-world objects annotated by their typical physical size. From this dataset, we selected 42 diverse objects spanning various categories and size ranges, which were subsequently mapped to a relatively uniform scale. The selection of the 42 objects was driven by a practical requirement, where the objects must be reliably identifiable by the underlying models in order to measure relational reasoning separately from recognition errors. To ensure this, we used in-context learning verification to confirm that both OpenFlamingo and Qwen3-VL could correctly name these objects.

Each image in the dataset has a resolution of $800 \times 800$ pixels and depicts six objects arranged to instantiate specific spatial relations. Four relations are considered: *above*, *below*, *left of*, and *right of*. One object is designated as the reference object, which consistently serves as the **query object** in the relational reasoning task. To maintain spatial centrality and leave room for neighboring objects, the reference object is randomly placed within a $400 \times 400$ central region of the image (bounded between pixels 200 and 600 along both axes). The four relational objects are then positioned directly above, below, left, and right of the reference object, corresponding to the four target spatial relations. Finally, a sixth object is placed at a minimum distance of 300 pixels from all other objects. Following this procedure, we used 32 object among the 42 objects and generated a total of 7000 images. These were divided into four subsets: (1) 4000 images for extracting function vectors, (2) 1000 images for fine-tuning function vectors, (3) 1000 images for evaluating generalization, and (4) 1000 images for a relation generalization test set containing four novel spatial relations not present in training: *above left*, *above right*, *below left*, and *below right*. The relation generalization test set was designed to support one-shot analogy tasks, enabling evaluation of the generalization capacity of multimodal function vectors to unseen relations.

In addition, we constructed an object generalization test set containing the held-out 10 objects not present in training. As in the main dataset, each image includes a centrally placed reference object, four relational objects corresponding to the target spatial relations, and one additional object positioned at least 300 pixels away from all others. In total, we generated 1000 such images for this test set.

### A.1.2  REAL IMAGE DATASET: GQA

For more realistic settings, we constructed a dataset using the GQA (Hudson & Manning, 2019), which consists of real-world images annotated with detailed scene graphs supporting visual reasoning and question answering. From the 113,000 images in GQA, we selected 4,226 images using strict criteria designed to target relational tasks. Specifically, (i) each object must have appeared only once per image, (ii) objects were required to occupy between 5% and 30% of the image area, (iii) non-descriptive or background-type objects (e.g., *sky, ground, tree, clothes, hair*) were removed, (iv) each image must contain between four and seven valid objects, (v) only seven designated spatial relations were considered (above, below, to the left of, to the right of, next to, behind, in front of), and (vi) each image must include at least four valid spatial relations and three distinct relation types. These constraints ensured that the final set of images captured relational structures suitable for evaluating visual relational reasoning.

Because the number of real images is limited, we divided the dataset into two subsets: a training set, used for function vector extraction and fine-tuning, and a test set, used exclusively for evaluation with zero-shot tasks. The 4,226 images were split in half while ensuring that the distribution of relation categories was balanced across the two subsets. The training set consists of 2,113 images with a total of 6,339 relation instances, and the test set consists of the other 2,113 images with 6,584 relation instances. Note that one image can include multiple spatial relations among objects.

We randomly sampled 1000 tasks for each relation in both the training and testing sets. Each task consists of four context images and one query image, all drawn from the corresponding set. In every

| Model | Synthetic (%) | | | GQA (%) | | |
|---|---|---|---|---|---|---|
| | 0-shot | 1-shot | 4-shot | 0-shot | 1-shot | 4-shot |
| Open-Flamingo | 4.8 | 8.7 | 9.7 | 11.8 | 16.8 | 19.0 |
| LLaVA-OneVision-1.5 | 10.7 | 12.8 | 17.7 | 8.0 | 18.2 | 25.5 |
| Qwen3-VL | 19.5 | 25.3 | 26.8 | 21.6 | 27.4 | 31.6 |

Table 1: In-context learning accuracy of three multimodal models on the Synthetic and GQA datasets across 0-shot, 1-shot, and 4-shot settings.

task, both the context and query images contain object pairs annotated with the target relation, and the specific object pairs instantiating the relation are randomly selected within each image.

### A.2 IN-CONTEXT LEARNING PERFORMANCE

In this section, we report the in-context learning capabilities of three large multimodal models on our two dataset under three prompting settings: zero-shot, one-shot, and four-shot. The models included in this comparison are Open-Flamingo-4B (Awadalla et al., 2023), LLaVA-OneVision-1.5-4B-Instruct (An et al., 2025; Liu et al., 2023a), and Qwen3-VL-4B-Instruct (Team, 2025; Bai et al., 2025).

Table 1 summarizes the prediction accuracy of each model across ICL settings. On both datasets, Qwen3-VL consistently achieves the highest accuracy in all three conditions, reflecting its stronger pretrained visual–language reasoning capabilities. LLaVA-OneVision shows moderate improvements with increasing numbers of in-context examples, while Open-Flamingo exhibits the weakest ICL performance overall, particularly in the synthetic setting. Across models, we observe a general upward trend from zero-shot to one-shot to four-shot prompting, indicating that exposure to task-relevant demonstrations helps all models to some extent.

### A.3 AVERAGE INDIRECT EFFECT

To quantify the overall contribution of an attention head in processing a specific relation, we compute its *average indirect effect* (AIE) as defined in Todd et al. (2024). This metric reflects the mean increase in the model's probability of generating the correct object label when the activation of attention head $a_{\ell j}$ is replaced by its relation-secific mean activations $\bar{a}_{\ell j}^t$ for perturbed prompts. The heads with the highest AIE scores are identified as the most causally influential for task execution and are grouped into the set $\mathcal{A}_t$.

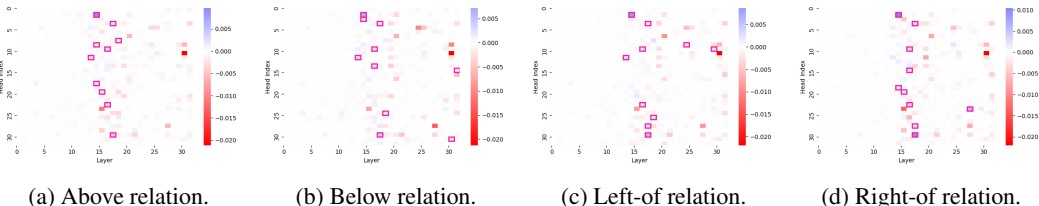

| (a) Above relation. | (b) Below relation. | (c) Left-of relation. | (d) Right-of relation. |

Figure 7: **AIE of attention heads for relations in the synthetic dataset.** Each heatmap shows the average indirect effect (AIE) values of attention heads (indexed by layer and head position). Pink boxes mark the top 10 most causally influential heads.

### A.4 ABLATION STUDIES

#### A.4.1 EFFECTS OF INJECTION LAYER

We examine how the effectiveness of function vector intervention varies across different injection layers. Zero-shot accuracy is evaluated when the vector is injected at each layer, while the base

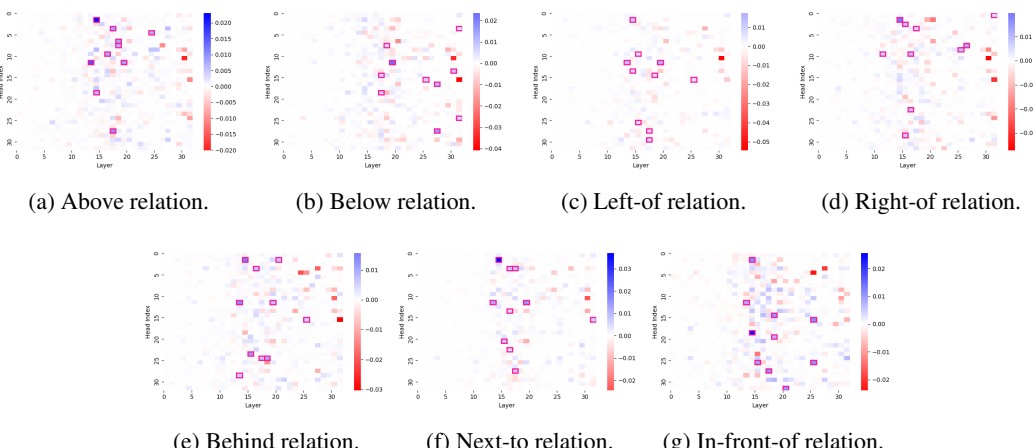

(a) Above relation.     (b) Below relation.     (c) Left-of relation.     (d) Right-of relation.

(e) Behind relation.     (f) Next-to relation.     (g) In-front-of relation.

Figure 8: **AIE of attention heads for relations in the real image dataset.** Each heatmap shows the average indirect effect (AIE) values of attention heads (indexed by layer and head position). Pink boxes mark the top 10 most causally influential heads.

model remains frozen and the intervention is applied only at the final token position of the query segment. As shown in Figure 9, zero-shot accuracy peaks when the function vector is injected at intermediate layers (around layer 19). Early-layer injection yields weaker effects due to limited semantic abstraction, whereas late-layer injection occurs too downstream to support structural reasoning. This non-monotonic pattern suggests that function vectors function not as linear modifiers but as triggers for nonlinear computations distributed across the model's depth.

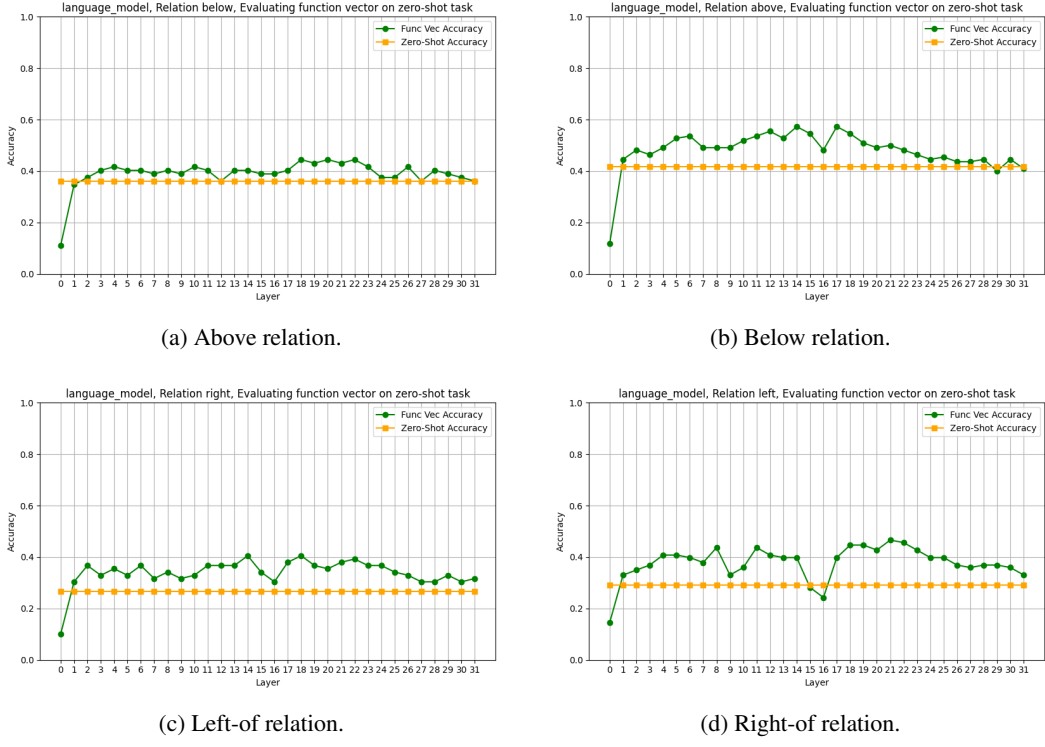

(a) Above relation.

(b) Below relation.

(c) Left-of relation.

(d) Right-of relation.

Figure 9: **Effect of injection layer on zero-shot accuracy.** Injecting the function vector at intermediate layers yields the highest accuracy, indicating that these layers are optimal for triggering relation computations.

### A.4.2 EFFECT OF HEAD SET SIZE IN FUNCTION VECTORS

We next analyze how the number of attention heads used to construct the function vector $\mathbf{v}_t$ influences zero-shot relational performance. We evaluate zero-shot accuracy as a function of $k \in \{1, 2, \ldots, 50\}$. Figure 10 presents the results for the relations in the synthetic dataset. In all cases, we observe a consistent non-monotonic trend: zero-shot accuracy improves rapidly as more top heads are included, reaches a peak in the range of 6 to 12 heads, and then gradually declines as additional, less informative heads are added.

This pattern highlights a trade-off: using too few attention heads underrepresent relational knowledge, while using too many attention heads introduces idiosyncratic activations from those with low or no causal relevance to spatial relations. Notably, for both relation types, the function vector significantly outperforms the unmodified zero-shot baseline when constructed from a small, carefully selected subset of heads. These findings reinforce the idea that relational reasoning is driven by a sparse set of causally influential attention heads.

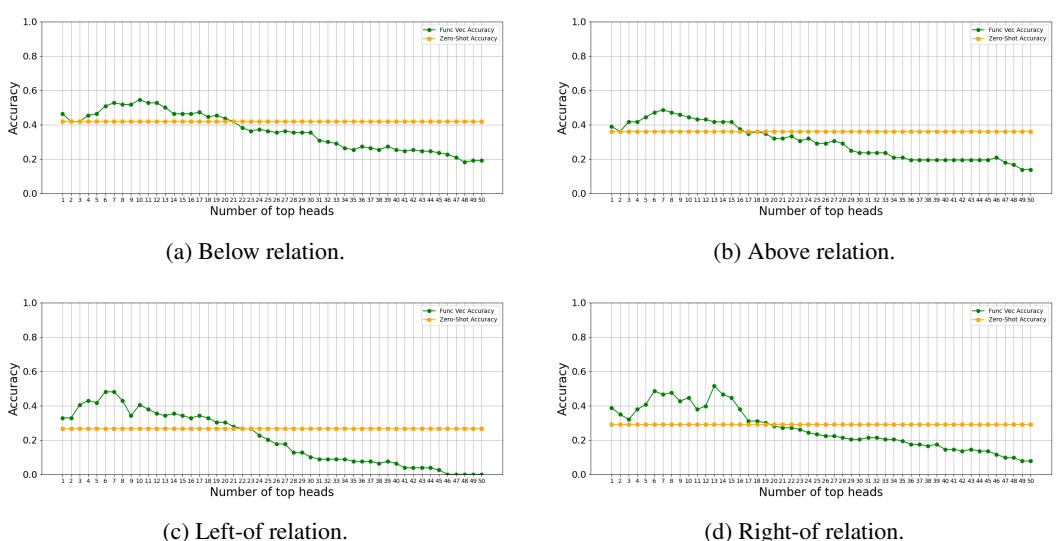

(a) Below relation.  (b) Above relation.

(c) Left-of relation.  (d) Right-of relation.

Figure 10: **Zero-shot accuracy as a function of number of heads in function vector.** Accuracy peaks when using 6 - 14 heads, suggesting that the function is distributed sparsely across a limited causal subnetwork.

### A.4.3 EFFECT OF CONTEXT SIZE DURING EXTRACTION

We analyze how the number of in-context examples used to extract head activations affects the performance of the relation-specific function vector $\mathbf{v}_t$. We vary the context size $n \in \{2, 4, 8\}$ used to construct ICL prompts when computing the task-conditioned head activations $\bar{a}^t_{\ell j}$, and evaluate zero-shot accuracy across layers.

Figure 11 presents results for the below and left of relations from synthetic dataset. Overall, we find that function vector performance is not highly sensitive to the number of context examples used during extraction. Accuracy remains relatively stable across 2-shot and 4-shot settings, especially in the middle layers where function vectors are most effective.

Interestingly, increasing the number of context examples beyond a moderate size does not necessarily yield better performance. In some cases, accuracy slightly declines when using 8-shot prompts compared to 4-shot. One possible explanation is that the relatively small size of the OpenFlamingo-4B model may limit its ability to integrate longer contexts effectively. This suggests that while some context is necessary to obtain stable and representative activations, more is not always better for LMMs.

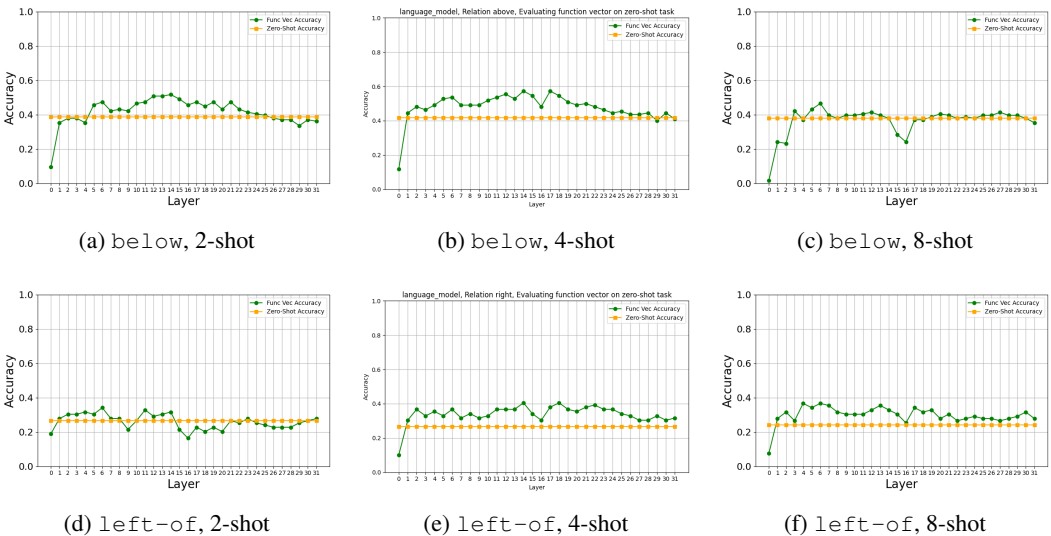

(a) below, 2-shot          (b) below, 4-shot          (c) below, 8-shot

(d) left-of, 2-shot        (e) left-of, 4-shot        (f) left-of, 8-shot

Figure 11: **Function vector accuracy across layers as a function of context size.** Each subfigure shows accuracy when injecting function vectors extracted from prompts with 2, 4, or 8 in-context examples. Results are shown for the below relation (top row) and left-of relation (bottom row).

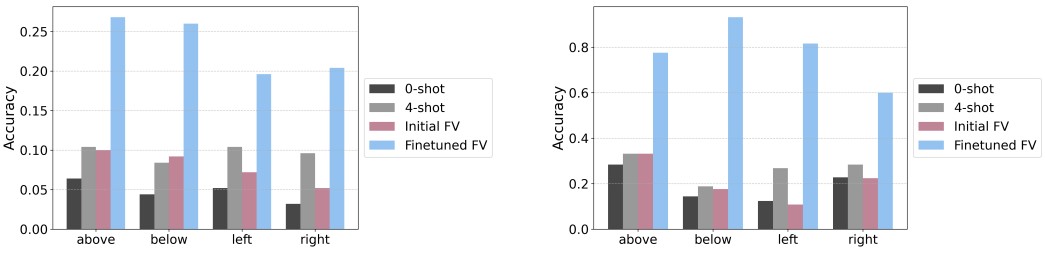

Figure 12: Per-relation top-1 prediction accuracy of zero-shot relation tasks for the OpenFlamingo model (left panel) and the Qwen3-VL model (right panel) on the Synthetic dataset. Finetuned vectors significantly outperform all baselines on the held-out zero-shot test set.

### A.5 MORE FUNCTION VECTORS FOR ZERO-SHOT RELATION TASK RESULTS

In this section, we provide the complete per-relation results corresponding to the averaged accuracies reported in the main text (Figure 5). Figure 12 present the detailed top-1 prediction accuracy for each individual relation in the zero-shot evaluation setting.

For the synthetic dataset, we report accuracies for all four spatial relations (*above*, *below*, *left-of*, *right-of*) after applying the initial relation-specific function vectors and their finetuned counterparts. As shown in Figure 12, finetuned function vectors (FFVs) consistently improve performance across all relations for both OpenFlamingo (left panel) and Qwen3-VL (right panel).

### A.6 COMPOSITE FUNCTION VECTORS FOR QWEN3-VL

We evaluate composite function vectors (CFVs) for one-shot analogy tasks using the Qwen3-VL model. To construct the CFVs, we derive function vectors from four primary spatial relations (*above, below, left-of, right-of*) in the source analogy and combine them through weighted averaging. The resulting CFV is then applied to the target analogy during inference. We compare the model's CFV-based performance with three baselines: zero-shot LMM, one-shot in-context learning (ICL), and four-shot ICL. Ten-shot ICL results are omitted due to GPU memory limitations. As shown in

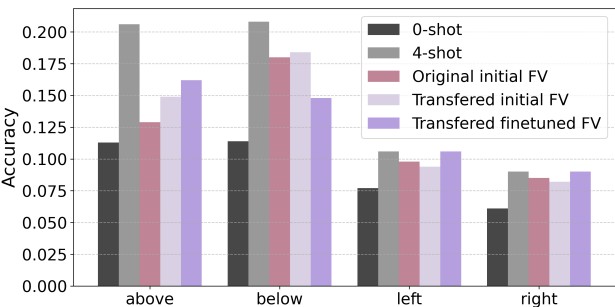

Figure 13: Transfer performance of function vectors from the Synthetic dataset to the GQA dataset for the OpenFlamingo model under five conditions: zero-shot, four-shot in-context learning, original GQA-derived function vectors, transferred (Synthetic-derived) initial function vectors, and transferred (Synthetic-)finetuned function vectors. Transferred function vectors achieve comparable or improved performance relative to GQA-derived function vectors and in-context learning, demonstrating the strong cross-dataset generalizability of function vectors.

Figure 3, Qwen3-VL with CFVs yields substantial gains, improving accuracy from 28.7% in the four-shot ICL setting to 45.1% with CFVs.

### A.7    TRANSFER FUNCTION VECTORS TO GQA DATASET

In this section, we evaluate the generalizability of function vectors by testing whether vectors learned on one dataset can transfer effectively to a different dataset with distinct features. Specifically, we extract relation-specific function vectors from the Synthetic dataset using the OpenFlamingo model and directly apply them to the GQA real-image dataset.

Figure 13 compares five conditions across four spatial relations: (i) the model's zero-shot baseline, (ii) four-shot in-context learning, (iii) the original function vectors obtained directly from GQA, (iv) the function vectors transferred from Synthetic dataset to GQA, and (v) the finetuned function vectors transferred from Synthetic dataset to GQA. We observe that the transferred function vectors achieve accuracy comparable to the original GQA-specific vectors across all relations, and in several cases exceed the performance of the original function vectors and the standard ICL prompting.

These results highlight that function vectors capture relational structure in a way that is robust across datasets and visual domains. Rather than overfitting to the synthetic environment, the learned vectors encode spatial relations in a generalizable form that transfers to naturalistic images. This cross-dataset transfer underscores the broader potential of function vectors as modular, reusable, and compositional components for reasoning in multimodal models.

### A.8    STRUCTURAL SIMILARITY OF FUNCTION VECTORS ACROSS MODELS

To assess the structural similarity of function vectors across OpenFlamingo and Qwen3-VL, we conducted a representational similarity analysis (RSA) over the four relation function vectors in the Synthetic dataset. The RSA correlations between the two models were negative for both the initial ($r = -0.38$) and finetuned function vectors ($r = -0.29$), indicating that the relational geometries encoded by the two models are not aligned. As shown in Table 2, OpenFlamingo's function vectors form a tightly clustered representation, with all pairwise cosine similarities above 0.9. In contrast, Table 3 shows that Qwen3-VL produces more differentiated function vectors, reflected in substantially lower pairwise similarities.

## B    ETHICS STATEMENT

This research complies with the ICLR Code of Ethics. The study did not involve human subjects, personally identifiable data, or sensitive information. The synthetic dataset was constructed using object cutouts from the Big and Small Objects dataset (Konkle & Oliva, 2012), which is publicly

Table 2: OpenFlamingo FV similarity matrix.

|  | Above | Below | Left | Right |
|---|---|---|---|---|
| Above | 1.00 | 0.94 | 0.90 | 0.94 |
| Below | 0.94 | 1.00 | 0.94 | 0.96 |
| Left | 0.90 | 0.94 | 1.00 | 0.93 |
| Right | 0.94 | 0.96 | 0.93 | 1.00 |

Table 3: Qwen3-VL FV similarity matrix.

|  | Above | Below | Left | Right |
|---|---|---|---|---|
| Above | 1.00 | 0.01 | 0.39 | 0.56 |
| Below | 0.01 | 1.00 | -0.02 | 0.13 |
| Left | 0.39 | -0.02 | 1.00 | 0.14 |
| Right | 0.56 | 0.13 | 0.14 | 1.00 |

available and licensed for research. The real-image dataset was derived from the publicly released GQA dataset (Hudson & Manning, 2019), and our subset selection followed criteria designed to preserve data integrity and avoid inclusion of sensitive or descriptive background elements. All datasets are used in accordance with their intended research purposes.

## C   REPRODUCIBILITY STATEMENT

A detailed description of dataset construction and preprocessing is provided in Supplementary Material A.1. All experimental settings, model architectures, and training procedures are reported in the main text. To further support reproducibility, we provide the full source code and all datasets used in our experiments through an anonymous OSF repository.

## D   USE OF LARGE LANGUAGE MODELS

Large language models (ChatGPT and Claude) were used as assistive tools for writing polish, generating plotting scripts for visualizing results, and debugging code. In addition, Figure 1 was generated using Gemini 2.5 Flash Image (Nano Banana). These tools did not contribute to research ideation, experimental design, data analysis, or substantive writing of the paper. All research ideas, experiments, and results were conceived and validated by the authors, who take full responsibility for the final content.

