# OpenReview forum: "Multimodal Function Vectors for Spatial Relations"
_ICLR.cc/2026/Conference — Submitted to ICLR 2026_

### Official Review · Reviewer_4SVg · 2025-10-28

**Soundness:** 2
**Presentation:** 3
**Contribution:** 2
**Rating:** 4
**Confidence:** 4

**Summary:**

The paper explore whether function vectors (FVs), that is activation of a sparse set of attention heads, can be extended from LLMs to LMMs for spatial relation reasoning. Using causal mediation analysis in OpenFlamingo-4B, the authors identify causally influential heads for each relation, sum their relation conditioned mean activations to form function vector. They further fine-tune only the function vector (freezing the LMM) to outperform few-shot ICL, and show that linear combinations of relation-specific function vector solve one-shot analogy tasks involving untrained composite relations. Experiments use both a purpose-built synthetic dataset and a carefully filtered GQA subset.

**Strengths:**

1. The work moves beyond heuristic head selection by using AIE/CIE to identify heads whose activations cause better relational predictions, then aggregates them into relation-specific FVs—bringing mechanistic interpretability tools to LMMs.
2. Merely injecting FVs improves zero-shot relation prediction; fine-tuning the FVs alone (freezing the model) significantly surpasses few-shot ICL, showing a compact, optimizable control handle for relational knowledge.
3. The synthetic set isolates relations; the GQA subset is stringently filtered to target relational reasoning, helping attribute gains to relation knowledge rather than object/category priors.

**Weaknesses:**

1. Results are limited to OpenFlamingo-4B; it remains unclear whether the same causal structure and FV utility hold for larger or different fusion architectures. The authors acknowledge this.
2. Only a few spatial relations are considered, and the GQA subset is small (201 images), limiting statistical power and real-world coverage.
3. Injecting or tuning FVs may perturb non-relational abilities (VQA, captioning, text understanding). No retained-capability analysis is reported.

**Questions:**

When injecting or fine-tuning the relation-specific FVs, what is the impact on other capabilities? Please report retained-capability results (e.g., standard VQA/captioning/text tasks) before vs. after FV injection and after FV fine-tuning.

---

> ### Author Response · Authors · 2025-11-20
> **Response to Reviewer 4SVg: Generalization to Stronger LMMs, Larger GQA Evaluation, and Impact on Non-Relational Abilities**
>
> We thank Reviewer 4SVg for the helpful and constructive comments.
>
> To address the concern about model scale and architectural generality, we conducted additional experiments on a more recent and more capable vision–language model with a different fusion architecture, Qwen3-VL-4B-Instruct. The qualitative patterns identified in our original function vector analysis were reproduced in Qwen3-VL, as detailed under section 4.3 in our revised manuscript. Moreover, the finetuned function vector (FFV) on Qwen3-VL yielded a large performance gain on the synthetic dataset, improving accuracy from 21% (initial FV) to 78.1%. These results indicate that the FV mechanism is not restricted to a single small model but extends to more powerful contemporary LMMs.
>
> Point-to-point responses:
> Weakness 1: We have added the new results from Qwen3-VL-4B, confirming that the proposed framework remains effective for a more modern vision–language model.
>
> Weakness 2: We substantially expanded the number of GQA images used, increasing the dataset from 201 images to more than 4,000. Notably, adding more images improved the performance of the function vectors, likely because the larger dataset produced more robust estimates of the average activations for key attention heads.
>
> Weakness 3/Question: Our paper specifically aims to investigate whether relational representations exist in large multimodal models and whether these representations can be enhanced through fine-tuning. For this reason, our method distills a function vector (FV) from the mean activations of key components and fine-tunes only this vector, which is external to the model parameters. The underlying model weights remain completely frozen throughout. Consequently, FV injection or FV fine-tuning does not modify the model’s parameters and therefore should not affect its general abilities on unrelated tasks such as standard VQA, captioning, or text understanding. While a retained-capability analysis would be valuable for future work, the design of our method operates entirely outside the model weights and therefore minimizes the risk of perturbing non-relational capabilities.

---

### Official Review · Reviewer_xGtF · 2025-10-30

**Soundness:** 3
**Presentation:** 3
**Contribution:** 2
**Rating:** 4
**Confidence:** 3

**Summary:**

This paper investigates how multimodal function vectors can influence the spatial reasoning capabilities of large multimodal models (LMMs). Building on the idea of "function vectors" from LLM interpretability research, the authors identify a small subset of attention heads in OpenFlamingo-4B responsible for encoding spatial relations. By extracting and manipulating the activations of these heads, they can modify the model’s performance on relational reasoning tasks.

The study uses causal mediation analysis, experiments on both synthetic and real-world datasets (e.g., GQA), and explores how these vectors can be fine-tuned or linearly combined to achieve compositional generalization. The results suggest that spatial relational knowledge in LMMs is modular, interpretable, and can be systematically controlled.

**Strengths:**

* The paper provides a novel and insightful exploration of how function vectors affect spatial perception in multimodal large models, extending interpretability research from LLMs to LMMs.

* Experimental results on both synthetic datasets and GQA show consistent improvement, demonstrating that function vectors can effectively manipulate spatial reasoning capabilities.

* The layer-wise and attention-head analyses are methodically performed, supporting the claimed relationship between specific attention structures and spatial reasoning.

* The composite function vector experiments exhibit a promising degree of generalization and compositionality, contributing new understanding to explainable and controllable AI.

**Weaknesses:**

1. The study is conducted only on OpenFlamingo-4B, which limits generalizability. Given the rapid progress in multimodal modeling, it would strengthen the paper to evaluate newer baselines such as Qwen-VL-2.5 (3B) or Gemma-3 (4B).

2. The selection of 32 object categories in the synthetic dataset is not well-justified. Some categories may appear ambiguous or prone to recognition errors by LMMs? Including more diverse or unseen object classes (in validation set) could better support the robustness of function vectors.

3. In the composite function vector experiments, the paper lacks zero-shot results or a discussion of correlation significance. This omission leaves unclear how strongly these vectors generalize without fine-tuning.

**Questions:**

1. Could you comment on how the identified function vectors might transfer across different LMM architectures?

2. What criteria guided the selection of object categories in the synthetic dataset?

3. Why are zero-shot results omitted in the composite vector experiments—were the correlations too weak or inconsistent?

4. Do you plan to test this method on stronger multimodal baselines to verify universality?

If these additional experiments or clarifications can be provided, I would be willing to raise my overall score, as the paper’s core idea is interesting and potentially impactful.

---

> ### Author Response · Authors · 2025-11-20
> **Response to Reviewer xGtF: Model Generalization, Object Selection, and Composite Relation Evaluation:**
>
> We appreciate Reviewer xGtF’s insightful comments and constructive suggestions. We agree that demonstrating generalizability beyond OpenFlamingo is important. To address this, we conducted additional experiments on a more recent vision–language model, Qwen3-VL-4B-Instruct, which is substantially stronger than OpenFlamingo in both visual understanding and spatial reasoning. As now reported in Section 4.3, all qualitative patterns from our original analysis replicated in Qwen3-VL. Moreover, the finetuned function vector (FFV) on Qwen3-VL led to a large performance gain, improving accuracy from 21.0% (initial FV) to 78.1% (finetuned FV). These findings provide preliminary but compelling evidence that the proposed mechanism extends to more powerful contemporary LMMs.
>
> Point-to-point responses:
> Weakness 1: We added the results from Qwen3-VL, which confirm the framework is still effective for a more recent vision-language model.
>
> Weakness 2: Our selection of 32 object categories was driven by a practical requirement, where the objects must be reliably identifiable by the underlying models in order to measure relational reasoning separately from recognition errors. To ensure this, we used in-context learning verification to confirm that both OpenFlamingo and Qwen3-VL could correctly name these objects. We have clarified this under Appendix A.1.1 in our revised manuscript.
> To further evaluate generalizability, we constructed a new test set containing 10 completely unseen object categories (not used during finetuning). As reported in Section 4.3, Qwen3-VL with finetuned function vectors reached 72.2% accuracy on this unseen-object set, compared to 23.3% for the initial FV and 21.3% for the baseline. These results indicate that function vectors do not overfit to specific visual categories, but instead encode relational structure in a manner that transfers robustly to novel objects.
>
> Weakness 3: The composite function vector (CFV) setting corresponds to reasoning about novel spatial relations (e.g., above-right) for which no function vector exists. To construct a CFV, the model must infer the appropriate mixture weights from a single source analogy example. Importantly, the source image is never provided as an input to the model during inference and it is only used to compute weights. Thus, although we refer to it as a “one-shot analogy task,” this is in fact strictly harder than zero-shot prompting, since the model must solve a weighting problem and then a no zero-shot counterpart.
> While one could manually supply analytic weights (e.g., above_right = 0.5 * above + 0.5 * right), doing so removes the analogy component of the task and artificially boosts performance. For this reason, we report only the analogy-based CFVs, which preserve the intended evaluation of compositional generalization. We clarify this in the revision under section 3.3.5.
>
> Question 1: We added a new analysis in Appendix A.8, where we directly compare the relational structure encoded by function vectors across OpenFlamingo and Qwen3-VL using representational similarity analysis (RSA). Specifically, the RSA correlations between the two models were negative for both the initial (r = −0.38) and finetuned function vectors (r = −0.29), indicating that the relational geometries encoded by the two models are not aligned.
> Despite these structural differences, Appendix A.7 shows that function vectors can transfer across datasets for the same model. Specifically, function vectors learned on the Synthetic dataset transfer effectively to the real-image GQA dataset, achieving accuracy comparable to GQA-derived function vectors. Thus, while architectures differ geometrically, the functional mechanism underlying relation manipulation is broadly applicable.
>
> Question 2: See the reply to weakness 2.
>
> Question 3: See the reply to weakness 3.
>
> Question 4: Yes. As described in Weakness 1, we have already conducted experiments on Qwen3-VL-4B-Instruct, a considerably stronger and more recent 4B-scale LMM.
>
> We thank you again for the valuable comments and for helping us clarify and improve the work.

---

> > ### Comment · Reviewer_xGtF · 2025-11-21
> >
> > Thanks for your comprehensive response and additional results, which has addressed most of my concerns.
> >
> > As for the zero-shot CFV experiment, I believe this leaves a promising venue for future works.
> >
> > I will raise my overall rating to 6.

---

> > > ### Author Response · Authors · 2025-11-21
> > >
> > > Thank you for your thoughtful feedback and for taking the time to review our work. We’re glad to hear that the additional results helped address your concerns. We also appreciate your perspective on the zero-shot CFV experiment and agree that it opens up interesting directions for future research.
> > >
> > > Thank you as well for updating your rating.

---

### Official Review · Reviewer_zkZ7 · 2025-10-31

**Soundness:** 2
**Presentation:** 2
**Contribution:** 2
**Rating:** 2
**Confidence:** 4

**Summary:**

This paper investigates whether the concept of "function vectors" (FVs), previously explored in large language models, can be extended to large multimodal models (LMMs) for the specific task of spatial reasoning. The authors use OpenFlamingo-4B and employ causal mediation analysis to identify a small subset of attention heads responsible for encoding spatial relations.

**Strengths:**

The experimental design is logical and methodical, progressing from identifying causally relevant components to extraction, fine-tuning, and testing for generalization.

The work extends a interesting line of research from LLMs to the multimodal domain, which is a potenial new direction for the community.

**Weaknesses:**

My primary concern is that the entire empirical investigation rests on a single, relatively dated and small-scale architecture and highly controlled, simplified datasets, as authors identified. Maybe the simplified spatial relation reasoning can be well-solved by latest VL model.

The evidence for strong generalization is not convincing because the analogy task is too simple. The "untrained" relations like "above-left" are just trivial combinations of the trained ones ("above" and "left"). The success of this linear combination is predictable for simple geometric directions and doesn't prove the method works for truly complex spatial relations like "inside," "leaning against," or "occluding." The experiment only validates the approach on an overly simplistic, best-case scenario.

The evaluation and comparsion seems very weak.

**Questions:**

Can you provide a strong argument for why you believe these findings (e.g., the specific layers and sparsity of influential heads) would transfer to more powerful LMMs like Qwen 2.5-VL models? Did you perform any preliminary experiments to suggest this is the case?

---

> ### Author Response · Authors · 2025-11-20
> **Response to Reviewer zkZ7: Transfer to Stronger LMMs, Generalization to Novel Objects & Dataset, and Task Complexity**
>
> We thank Reviewer zkZ7 for the constructive and insightful feedback.
>
> To assess whether our findings generalize beyond the dated OpenFlamingo model used in the paper, we conducted additional experiments on two more recent and more powerful vision–language model, LLaVA-OneVision-1.5-4B-Instruct and Qwen3-VL-4B-Instruct. We added in-context learning performance for all models under Appendix A.2 in our revised manuscript. Under 4-shot in-context learning,  LLaVA-OneVision achieved 17.7% while Qwen3-VL achieved 26.8% accuracy on our synthetic dataset, substantially higher than OpenFlamingo, but still far from solving the task. Importantly, the qualitative patterns identified in our function vector analysis were reproduced in Qwen3-VL as well, as detailed under section 4.3 in our revised manuscript. Moreover, the finetuned function vector (FFV) led to a large performance gain on Qwen3-VL, improving accuracy from 21.0% (initial FV) to 78.1% (finetuned FV). These findings provide preliminary but compelling evidence that the proposed mechanism extends to more powerful contemporary LMMs.
>
> To further address the question of generalization, we constructed a new test set containing 10 novel objects never seen during fine-tuning. On this evaluation, Qwen3-VL with fine-tuned FVs again produced substantial gains, reaching 72.2% accuracy, compared to 23.3% for the initial FV and 21.3% for the baseline. These results, now included in Section 4.3 of the revised manuscript, demonstrate that the relational knowledge encoded in function vectors generalizes to unseen objects. We additionally added a new analysis in Appendix A.7 showing that function vectors can transfer across datasets: FVs learned on the synthetic dataset transfer effectively to the real-image GQA dataset, achieving accuracy comparable to function vectors trained directly on GQA. These results provide converging evidence that function vectors capture relational structure that is robust and broadly generalizable across both objects and datasets.
>
> Finally, regarding the concern that the CFV was evaluated only on “too simple” spatial relations, we emphasize that this work is intended as a proof of concept. Notably, even the more capable Qwen3-VL model attains relatively low in-context learning accuracy on this task, indicating that the setting remains challenging for contemporary LMMs. Extending the method to a broader and more complex set of spatial relations is an important direction for future research, and we acknowledge this limitation explicitly.

---

### Author Response · Authors · 2025-12-02
**Author Response Summary for the Area Chair**

Across all three reviewers, the core idea of function vectors (FVs) in LMMs was recognized as interesting, novel, and potentially impactful. Reviewers highlighted the following strengths:

1. Clear methodology built upon causal mediation analysis.
2. Insightful extension of mechanistic interpretability from LLMs to multimodal models.
3. Evidence that sparse attention-head–based representations can systematically influence spatial reasoning.
4. Promising compositionality via composite FVs.
5. Carefully constructed datasets for isolating spatial relations.

Below, we summarize the main reviewer concerns and our consolidated responses:

1. All initial results were based on OpenFlamingo-4B and it is unclear whether findings extend to stronger LMMs. (all reviewers)

Our response: We extended our analysis to the more capable Qwen3-VL-4B-Instruct. The qualitative interpretability patterns observed in OpenFlamingo replicate in Qwen3-VL, and finetuned FVs remain highly effective, improving performance from 21% to 78.1%. These results provide direct evidence that the FV mechanism generalizes to modern architectures.

2. Generalization to unseen objects, and GQA evaluation was small. (Reviewer xGtF, 4SVg)

Our response: We introduced a new synthetic test set containing 10 previously unseen object categories, and Qwen3-VL with finetuned FVs still achieved 72.2% accuracy, showing that FVs capture abstract relational structure rather than object-specific cues.
Additionally, Appendix A.7 now includes new results demonstrating that FVs trained on the synthetic dataset transfer effectively to GQA, performing comparably to FVs directly trained on GQA.
To strengthen the empirical basis, we expanded the curated GQA subset from 201 images to over 4,000, improving statistical power and robustness.

3. The composite function vector task is too simple and there is no zero-shot or baseline comparisons for composite function vectors (Reviewer zkZ7, xGtF)

Our response: We clarify that the CFV setting is strictly harder than zero-shot: the model must infer mixture weights from a single analogy example. Providing analytic weights (e.g., averaging “above” and “right”) artificially boosts performance and removes the analogy component. Section 3.3.5 now explains why zero-shot CFVs do not meaningfully evaluate compositional generalization.

4. The task may be too simple, and recent VL models might already solve it without specialized intervention. (Reviewer zkZ7)

Our response: To directly evaluate this possibility, we added results from two stronger, modern VLMs, LLaVA-OneVision-1.5-4B-Instruct and Qwen3-VL-4B-Instruct. Despite their substantial improvements in general visual understanding, both models still perform far below satisfactory levels on our controlled spatial-relation benchmark (e.g., 17.7% and 26.8% accuracy under 4-shot ICL), indicating that the task is not yet solved even by state-of-the-art systems.

5. Does FV injection or tuning degrade unrelated abilities? (Reviewer 4SVg)

Our response: Our method does not modify model parameters. FVs operate by adding an external vector to a subset of hidden activations, while all LMM weights remain frozen. Because no parameters are changed, unrelated abilities are not expected to be affected.
These findings underscore that fine-grained spatial reasoning remains a challenging open problem for current VLMs, and that our benchmark provides a valuable new evaluation setting for probing relational understanding in multimodal models.

Finally, we note that Reviewer xGtF stated on Nov 21 (before the identity leak incident) that they would like to raise their score to 6, indicating that the clarifications and new experiments satisfactorily addressed their concerns.

---

### Meta-Review · Area_Chair_AsgN · 2026-01-07

**Summary:**

The paper investigates whether function vectors—previously studied primarily in LLMs—can be extended to multimodal LMMs for the task of spatial reasoning. While the reviewers unanimously found the problem interesting and potentially impactful, they noted that the experimental design could be substantially strengthened by evaluating larger, state-of-the-art models, incorporating more challenging datasets, and comparing against stronger baselines. During the discussion period, the authors provided additional experimental results, which partially addressed these points. However, further empirical validation is needed to fully support the claims, and the required changes would likely necessitate a major revision. After extensive discussion, the AC finds that the weaknesses outweigh the strengths and therefore recommends rejection. The authors are encouraged to incorporate the reviewers’ feedback and consider submitting to a future venue.

**Reviewer Concerns:**

See above.

**Reviewer Scores:**

See above.

---

### Decision · Program_Chairs · 2026-01-26

Reject